# WaNet – Imperceptible Warping-based Backdoor Attack

**Anh Tuan Nguyen**[1,2]**, Anh Tuan Tran**[1,3]
[1]VinAI Research, [2]Hanoi University of Science and Technology, [3]VinUniversity
{v.anhnt479,v.anhtt152}@vinai.io

## Abstract

With the thriving of deep learning and the widespread practice of using pre-trained networks, backdoor attacks have become an increasing security threat drawing many research interests in recent years. A third-party model can be poisoned in training to work well in normal conditions but behave maliciously when a trigger pattern appears. However, the existing backdoor attacks are all built on noise perturbation triggers, making them noticeable to humans. In this paper, we instead propose using warping-based triggers. The proposed backdoor outperforms the previous methods in a human inspection test by a wide margin, proving its stealthiness. To make such models undetectable by machine defenders, we propose a novel training mode, called the "noise" mode. The trained networks successfully attack and bypass the state of the art defense methods on standard classification datasets, including MNIST, CIFAR-10, GTSRB, and CelebA. Behavior analyses show that our backdoors are transparent to network inspection, further proving this novel attack mechanism's efficiency. Our code is publicly available at https://github.com/VinAIResearch/Warping-based_Backdoor_Attack-release.

## 1 Introduction

Deep learning models are essential in many modern systems due to their superior performance compared to classical methods. Most state-of-the-art models, however, require expensive hardware, huge training data, and long training time. Hence, instead of training the models from scratch, it is a common practice to use pre-trained networks provided by third-parties these days. This poses a serious security threat of backdoor attack (Gu et al., 2017). A backdoor model is a network poisoned either at training or finetuning. It can work as a genuine model in the normal condition. However, when a specific trigger appears in the input, the model will act maliciously, as designed by the attacker. Backdoor attack can occur in various tasks, including image recognition (Chen et al., 2017), speech recognition (Liu et al., 2018b), natural language processing (Dai et al., 2019), and reinforcement learning (Hamon et al., 2020). In this paper, we will focus on image classification, the most popular attacking target with possible fatal consequences (e.g., for self-driving car).

Since introduced, backdoor attack has drawn a lot of research interests (Chen et al., 2017; Liu et al., 2018b; Salem et al., 2020; Nguyen & Tran, 2020). In most of these works, trigger patterns are based on patch perturbation or image blending. Recent papers have proposed novel patterns such as sinusoidal strips (Barni et al., 2019), and reflectance (Liu et al., 2020). These backdoor triggers, however, are unnatural and can be easily spotted by humans.

We believe that the added content, such as noise, strips, or reflectance, causes the backdoor samples generated by the previous methods strikingly detectable. Instead, we propose to use image warping that can deform but preserve image content. We also found that humans are not good at recognizing subtle image warping, while machines are excellent in this task.

Hence, in this paper, we design a novel, simple, but effective backdoor attack based on image warping called WaNet. We use a small and smooth warping field in generating backdoor images, making the modification unnoticeable, as illustrated in Fig. 1. Our backdoor images are natural and hard to be distinguished from the genuine examples, confirmed by our user study described in Sec. 4.3.

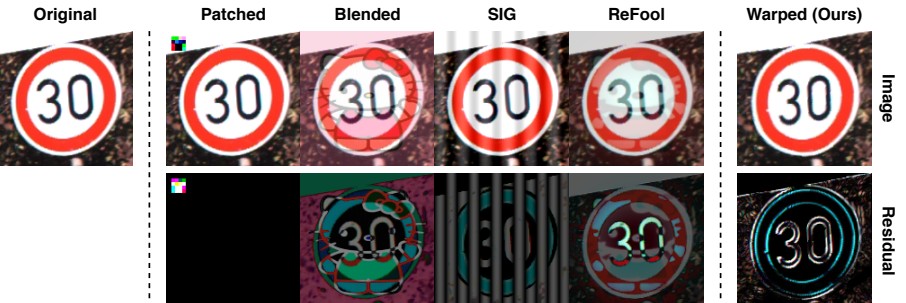

Figure 1: **Comparison between backdoor examples generated by our method and by the previous backdoor attacks.** Given the original image (leftmost), we generate the corresponding backdoor images using patch-based attacks (Gu et al., 2017; Liu et al., 2018b), blending-based attack (Chen et al., 2017), SIG (Barni et al., 2019), ReFool (Liu et al., 2020), and our method. For each method, we show the image (top), the magnified ($\times 2$) residual map (bottom). The images generated from the previous attacks are unnatural and can be detected by humans. In constrast, ours is almost identical to the original image, and the difference is unnoticeable.

To obtain a backdoor model, we first follow the common training procedure by poisoning a part of training data with a fixed ratio of $\rho_a \in (0, 1)$. While the trained networks provide high clean and attack accuracy, we found that they "cheated" by learning pixel-wise artifacts instead of the warping itself. It makes them easy to be caught by a popular backdoor defense Neural Cleanse. Instead, we add another mode in training, called "noise mode", to enforce the models to learn only the predefined backdoor warp. This novel training scheme produces satisfactory models that are both effective and stealthy.

Our attack method achieves invisibility without sacrificing accuracy. It performs similarly to state-of-the-art backdoor methods in terms of clean and attack accuracy, verified on common benchmarks such as MNIST, CIFAR-10, GTSRB, and CelebA. Our attack is also undetectable by various backdoor defense mechanisms; none of existing algorithms can recognize or mitigate our backdoor. This is because the attack mechanism of our method is drastically different from any existing attack, breaking the assumptions of all defense methods.

Finally, we demonstrate that our novel backdoor can be a practical threat by deploying it for physical attacks. We tested the backdoor classifier with camera-captured images of physical screens. Despite image quality degradation via extreme capturing conditions, our backdoor is well-preserved, and the attack accuracy stays near 100%.

In short, we introduce a novel backdoor attack via image warping. To train such a model, we extend the standard backdoor training scheme by introducing a "noise" training mode. The attack is effective, and the backdoor is imperceptible by both humans and computational defense mechanisms. It can be deployed for physical attacks, creating a practical threat to deep-learning-based systems [1].

## 2 BACKGROUND

### 2.1 THREAT MODEL

Backdoor attacks are techniques of poisoning a system to have a hidden destructive functionality. The poisoned system can work genuinely on clean inputs but misbehave when a specific trigger pattern appears. In the attack mode for image classification, backdoor models can return a predefined target label, normally incorrect, regardless of image content. It allows the attacker to gain illegal benefits. For example, a backdoor face authentication system may allow the attacker to access whenever he puts a specific sticker on the face.

Backdoors can be injected into the deep model at any stage. We consider model poisoning at training since it is the most used threat model. The attacker has total control over the training process and maliciously alters data for his attack purposes. The poisoned model is then delivered to customers to

---

[1]Source code of the experiments will be publicly available.

deploy as-is. In our proposed attack, the attacker selects a fixed warping field and uses it to generate all the backdoor images in training and in testing-time attacks.

## 2.2 PREVIOUS BACKDOOR ATTACKS

We focus on backdoor attacks on image classification. The target network is trained for a classification task $f : \mathbb{X} \to \mathbb{C}$, where $\mathbb{X}$ is an image domain and $\mathbb{C} = \{c_1, c_2, ..., c_M\}$ is a set of $M$ target classes. When poisoning $f$, we enforce it to learn an injection function $\mathcal{B}$, a target label function $c$, and alter the network behaviour so that:

$$f(\boldsymbol{x}) = y, \quad f(\mathcal{B}(\boldsymbol{x})) = c(y) \tag{1}$$

for any pair of clean image $\boldsymbol{x} \in \mathbb{X}$ and the corresponding label $y \in \mathbb{C}$.

The earliest backdoor attack was BadNets (Gu et al., 2017). The authors suggested to poison a portion of training data by replacing each clean data pair $(\boldsymbol{x}, y)$ with the corresponding poisoned pair $(\mathcal{B}(\boldsymbol{x}), c(y))$. The injection function $\mathcal{B}$ simply replaces a fixed patch of the input image by a predefined trigger pattern. As for the target label function $c(y)$, the authors proposed two tests: (1) **all-to-one** with a constant target label $c(y) = \hat{c}$ and (2) **all-to-all** with $c(y) = y + 1$.

After BadNets, many variants of backdoor attacks have been introduced. These approaches focus on changing either the backdoor injection process or the injection function $\mathcal{B}$.

As for the backdoor injection process, Liu et al. (2018b) proposed to inject backdoor into clean models via fine-tuning instead of the training stage. Yao et al. (2019) suggested hiding backdoor inside latent neurons for transfer learning. Many recent studies (Turner et al., 2019; Barni et al., 2019; Liu et al., 2020), injected backdoor only on samples with unchanged labels, i.e., the target $c(y)$ is the same as the ground-truth label $y$, to dodge label inspection by humans.

In this paper, we focus on the development of a good injection function $\mathcal{B}$. Most of the popular attack methods rely on fixed patch-based triggers. Chen et al. (2017) used image blending to embed the trigger into the input image, and Nguyen & Tran (2020) extended it to be input-aware. Salem et al. (2020) varied the patch-based trigger locations and patterns to make it "dynamic". Barni et al. (2019) employed sinusoidal strips as the trigger alongside the clean-label strategy. Lately, Liu et al. (2020) proposed to disguise backdoor triggers as reflectance to make the poisoned images look natural. The backdoor images generated by these attacks, however, are easy to be spotted by humans. We instead propose an "invisible" backdoor that is imperceptible by even sharp-eyed people.

## 2.3 BACKDOOR DEFENSE METHODS

As the threat of backdoor attacks becomes more apparent, backdoor defense research is emerging. Based on usage scenarios, we can classify them into three groups: training defense, model defense, and testing-time defense.

Training defense assumes the defender has control over the training process, and the adversary attacks by providing infected training data (Tran et al., 2018). This assumption, however, does not match our threat model, where the already-trained backdoor model is provided by a third party. This mechanism is not applicable to our situation and will not be considered further in this paper.

Model defenses aim to verify or mitigate the provided model before deployment. Fine-Pruning (Liu et al., 2018a) suggested to prune the dormant neurons, defined by analyses on a clean image set, to mitigate the backdoor if present. Neural Cleanse (Wang et al., 2019) was the first work that could detect backdoor models. It optimized a patch-based trigger candidate for each target label, then detected if any candidate was abnormally smaller than the others as a backdoor indicator. ABS (Liu et al., 2019) scanned the neurons and generated trigger candidates by reverse engineering. Cheng et al. (2019) used GradCam (Selvaraju et al., 2017) to analyze the network behavior on a clean input image with and without the synthesized trigger to detect anomalies. Zhao et al. (2019) applied mode connectivity to effectively mitigate backdoor while keeping acceptable performance. Lately, Kolouri et al. (2020) introduced universal litmus patterns that can be fed to the network to detect backdoor.

Unlike model defense, testing-time defenses inspect models after deployment with the presence of input images. It focuses on verifying if the provided image is poisoned and how to mitigate it. STRIP (Gao et al., 2019) exploited the persistent outcome of the backdoor image under perturbations

for detection. In contrast, Neo (Udeshi et al., 2019) searched for the candidate trigger patches where region blocking changed the predicted outputs. Recently, Doan et al. (2019) used GradCam inspection to detect potential backdoor locations. In all these methods, the trigger candidates were then verified by being injected into a set of clean images.

A common assumption in all previous defense methods is that the backdoor triggers are image patches. We instead propose a novel attack mechanism based on image warping, undermining the foundation of these methods.

## 2.4 ELASTIC IMAGE WARPING

Image warping is a basic image processing technique that deforms an image by applying the geometric transformation. The transformation can be affine, projective, elastic, or non-elastic. In this work, we propose to use elastic image warping given its advantages over the others: (1) Affine and projective transformations are naturally introduced to clean images via the image capturing process. If we apply these transformations to these images, the transformed images can be identical to other clean images that are of the same scenes but captured at different viewpoints. Hence, these transformations are not suitable to generate backdoor examples, particularly in physical attacks. (2) Elastic transformation still generates natural outputs while non-elastic one does not.

The most popular elastic warping technique is Thin-Plate Splines (TPS) (Duchon, 1977). TPS can interpolate a smooth warping field to transform the entire image given a set of control points with known original and target 2D coordinates. TPS was adopted in Spatial Transformer Networks (Jaderberg et al., 2015), the first deep learning study incorporating differential image warping.

We believe that elastic image warping can be utilized to generate invisible backdoor triggers. Unlike previous attack methods that introduce extra and independent information to an input image, elastic image warping only manipulates existing pixels of the image. Humans, while being excellent in spotting incongruent part of an image, are bad at recognizing small geometric transformations.

## 3 WARPING-BASED BACKDOOR ATTACK

We now describe our novel backdoor attack method WaNet, which stand for Warping-based poisoned Networks. WaNet are designed to be stealthy to both machine and human inspections.

### 3.1 OVERVIEW

Recall that a classification network is a function $f : \mathbb{X} \to \mathbb{C}$, in which $\mathbb{X}$ is an input image domain and $\mathbb{C}$ is a set of target classes. To train $f$, a training dataset $\mathbb{S} = \{(\boldsymbol{x}_i, y_i) | \boldsymbol{x}_i \in \mathbb{X}, y_i \in \mathbb{C}, i = \overline{1, N}\}$ is provided. We follow the training scheme of BadNets to poison a subset of $\mathbb{S}$ with ratio $\rho_a$ for backdoor training. Each clean pair $(\boldsymbol{x}, y)$ will be replaced by a backdoor pair $(\mathcal{B}(\boldsymbol{x}), c(y))$, in which $\mathcal{B}$ is the backdoor injection function and $c(y)$ is the target label function.

Our main focus is to redesign the injection function $\mathcal{B}$ based on image warping. We construct $\mathcal{B}$ using a warping function $\mathcal{W}$ and a predefined warping field $\boldsymbol{M}$:

$$\mathcal{B}(\boldsymbol{x}) = \mathcal{W}(\boldsymbol{x}, \boldsymbol{M}). \tag{2}$$

$\boldsymbol{M}$ acts like a motion field; it defines the *relative* sampling location of *backward* warping for each point in the target image. $\mathcal{W}$ allows a floating-point warping field as input. When a sampling pixel falls on non-integer 2D coordinates, it will be bi-linear interpolated. To implement $\mathcal{W}$, we rely on the public API $grid\_sample$ provided by PyTorch. However, this API inputs a grid of normalized *absolute* 2D coordinates of the sampling points. To use that API, we first sum $\boldsymbol{M}$ with an identity sampling grid, then normalize to $[-1, 1]$ to get the required grid input.

### 3.2 WARPING FIELD GENERATION

The warping field $\boldsymbol{M}$ is a crucial component; it must guarantee that the warped images are both natural and effective for attacking purposes. Hence, $\boldsymbol{M}$ are desired to satisfy the following properties:

- **Small**: $\boldsymbol{M}$ should be small, to be unnoticeable to humans,

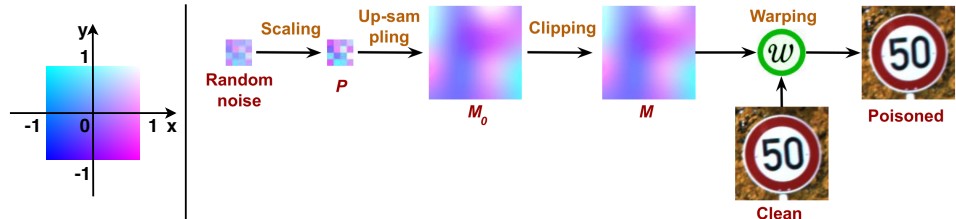

Figure 2: Process of creating the warping field $M$ and using it to generate poisoned images.

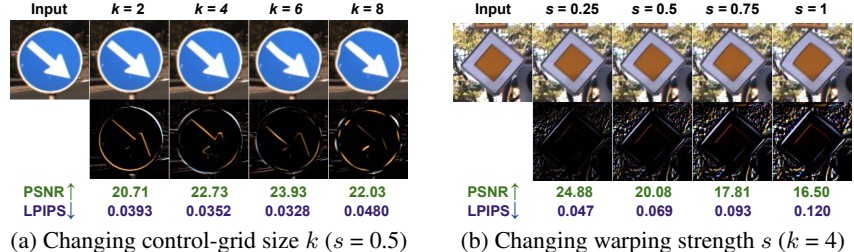

(a) Changing control-grid size $k$ ($s = 0.5$)  (b) Changing warping strength $s$ ($k = 4$)

Figure 3: **Effect of different hyper-parameters on the warping result.** For each warped image, we show the image (top), the magnified ($\times 2$) residual map (bottom). The PSNR and LPIPS (Zhang et al., 2018) scores are computed at resolution 224$\times$224.

- **Elastic**: $M$ should be elastic, i.e., smooth and non-flat, to generate natural looking images,
- **Within image boundary**: $M$ should not exceed the image boundary, to avoid creating suspicious black/plain outer area.

To get such a warping field, we borrow the idea of using control points from TPS but simplify the interpolation method. The process of generating the desired warp is illustrated by Fig. 2 and is described in the following subsections.

**Selecting the control grid** We first select the control points. For simplicity, we pick the target points on a uniform grid of size $k \times k$ over the entire image. Their backward warping field is denoted as $P \in \mathbb{R}^{k \times k \times 2}$. We use a parameter $s$ to define the strength of $P$ and generate $P$ as following:

$$P = \psi(rand_{[-1,1]}(k, k, 2)) \times s \tag{3}$$

in which $rand_{[-1,1]}(\dots)$ is a function returning random tensor with the input shape and element value in the range $[-1, 1]$ and $\psi$ is a normalization function. In this paper, we normalize the tensor elements by their mean absolute value:

$$\psi(A) = \frac{A}{\frac{1}{size(A)} \sum_{a_i \in A} |a_i|} \tag{4}$$

**Upsampling** From the control points, we interpolate the warping field of the entire image. Since these points are in a uniform grid covering the entire image, instead of using a complex spline-based interpolation like in TPS, we can simply apply bicubic interpolation. We denote the output of this step as $M_0 = \uparrow P \in \mathbb{R}^{h \times w \times 2}$, with $h$ and $w$ being the image height and width respectively.

**Clipping** Finally, we apply a clipping function $\phi$ so that the sampling points do not fall outside of the image border. The process of generating $M$ can be summarized by the equation:

$$M = \phi(\uparrow (\psi(rand_{[-1,1]}(k, k, 2)) \times s)). \tag{5}$$

We investigate the effect of the hyper-parameters $k$ and $s$ qualitatively in Fig. 3. The warping effect is almost invisible when $k < 6$ and $s < 0.75$.

### 3.3 RUNNING MODES

After computing the warping field $M$, we can train WaNet with with two modes, clean and attack, as the standard protocol. However, the models trained by that algorithm, while still achieving high

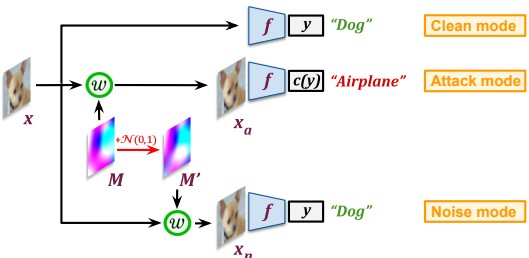

Figure 4: Training pipeline with three running modes.

| Dataset | Clean | Attack | Noise |
|---------|-------|--------|-------|
| MNIST | 99.52 | 99.86 | 98.20 |
| CIFAR-10 | 94.15 | 99.55 | 93.55 |
| GTSRB | 98.97 | 98.78 | 98.01 |
| CelebA | 78.99 | 99.33 | 76.74 |

(a) Network performance          (b) Sample backdoor images          (c) Physical attack test

Figure 5: Attack experiments. In (b), we provide the clean (top) and backdoor (bottom) images.

accuracy in both clean and attack tests, tend to learn pixel-level artifacts instead of the warping. They are, therefore, easily exposed by a backdoor defense method such as Neural Cleanse. We will discuss more details in the ablation studies in Section 4.6.

To resolve this problem, we propose a novel training mode alongside the clean and attack mode, called noise mode. The idea is simple: when applying a random warping field $M' \neq M$, the network should not trigger the backdoor but return the correct class prediction.

Fig. 4 illustrates three running modes in our training pipelines. We first select the backdoor probability $\rho_a \in (0,1)$ and the noise probability $\rho_n \in (0,1)$ such that $\rho_a + \rho_n < 1$. Then, for each clean input $(\boldsymbol{x}, y)$, we randomly select one of three modes and alter that pair accordingly:

$$(\boldsymbol{x}, y) \mapsto \begin{cases} (\boldsymbol{x}, y) & \text{with probability } 1 - \rho_a - \rho_n \\ (\mathcal{W}(\boldsymbol{x}, \boldsymbol{M}), c(y)) & \text{with probability } \rho_a \\ (\mathcal{W}(\boldsymbol{x}, \boldsymbol{M} + rand_{[-1,1]}(h, w, 2)), y) & \text{with probability } \rho_n \end{cases} \qquad (6)$$

Note that with the noise mode, instead of using a totally random warping field, we form it by adding Gaussian noise to $\boldsymbol{M}$ for a more effective training. The modified training set is then used to train $f$.

## 4 EXPERIMENTS

### 4.1 EXPERIMENTAL SETUP

Following the previous backdoor attack papers, we performed experiments on four datasets: MNIST (LeCun et al., 1998), CIFAR-10 (Krizhevsky et al., 2009), GTSRB (Stallkamp et al., 2012) and CelebA (Liu et al., 2015). Note that CelebA dataset has annotations for 40 independent binary attributes, which is not suitable for multi-class classification. Therefore, we follow the configuration suggested by Salem et al. (2020) to select the top three most balanced attributes, including Heavy Makeup, Mouth Slightly Open, and Smiling, then concatenate them to create eight classification classes. Their detail information are shown in Table 1. To build the classifier $f$ for the color image datasets, we used Pre-activation Resnet-18 (He et al., 2016) for the CIFAR-10 and GTSRB datasets as suggested by Kang (2020), and Resnet-18 for the CelebA dataset. As for the grayscale dataset MNIST, we defined a simple network structure as reported in Table 1.

We trained the networks using the SGD optimizer. The initial learning rate was 0.01, which was reduced by a factor of 10 after each 100 training epochs. The networks were trained until convergence. We used $k = 4$, $s = 0.5$, $\rho_a = 0.1$, and $\rho_n = 0.2$.

Table 1: Datasets and the classifiers used in our experiments. Each ConvBlock consists of a 3×3 convolution (stride=2), a BatchNorm, and a ReLU layer.

| Dataset | Subjects | #Classes | Input Size | #Train. Images | Classifier |
|---------|----------|----------|------------|----------------|------------|
| MNIST | Written digits | 10 | $28 \times 28 \times 1$ | 60,000 | 3 ConvBlocks, 2 fcs |
| CIFAR-10 | General objects | 10 | $32 \times 32 \times 3$ | 50,000 | PreActRes18 |
| GTSRB | Traffic signs | 43 | $32 \times 32 \times 3$ | 39,252 | PreActRes18 |
| CelebA | Face attributes | 8 | $64 \times 64 \times 3$ | 202,599 | ResNet18 |

| Fooling rate (%) | Patched | Blended | SIG | ReFool | WaNet |
|------------------|---------|---------|-----|--------|-------|
| Backdoor inputs | 8.7 | 1.4 | 2.7 | 2.3 | **38.6** |
| Clean inputs | 6.1 | 10.1 | 2.6 | 13.1 | **17.4** |
| All inputs | 7.4 | 5.7 | 2.6 | 7.7 | **28.0** |

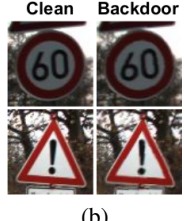

(a)                       (b)

Figure 6: **Human inspection tests:** (a) Success fooling rates of each backdoor method, (b) The most distinguishable cases from WaNet.

## 4.2 ATTACK EXPERIMENTS

We trained and tested the backdoor models in all-to-one configuration, i.e., $c(y) = \hat{c} \forall y$. The accuracy values in clean mode, attack mode, and the noise mode are reported in Fig. 5a. As can be seen, with clean images, the networks could correctly classify them like any benign models, with accuracy near 100% on MNIST/GTSRB, 94.15% on CIFAR-10, and 79.77% on CelebA. When applying the pre-defined image warping, the attack success rate was near 100% on all datasets. However, when using a random warping, the classifiers still recognized the true image class with a similar accuracy as in the clean mode. This result is impressive given the fact that the poisoned images look almost identical to the original, as can be seen in Fig. 5b.

To evaluate our method's robustness in real-life scenarios, we also tested if backdoor images would still be misclassified even when being distorted by the capturing process. We showed 50 clean and 50 backdoor images on a screen and recaptured them using a phone camera. Our model still worked well on recaptured images, obtaining 98% clean accuracy and 96% attack success rate. Fig. 5c displays an example of our test. The clean image was recognized correctly as "automobile", while the look-a-like backdoor image was recognized as the "airplane" attack class.

## 4.3 HUMAN INSPECTION

To examine the realisticity of our backdoor and the previous methods, we created user studies with human inspection. First, we randomly selected 25 images from the GTSRB dataset. Second, for each backdoor injection function, we created the corresponding 25 backdoor images and mixed them with the original to obtain a set of 50 images. Finally, we asked 40 people to classify whether each image was genuine, collecting 2000 answers per method. The participants were trained about the mechanism and characteristics of the attack before answering the questions.

We collected the answers and reported the percentage of incorrect answers as the success fooling rates in Fig. 6a. Note that when the backdoor examples are more indistinguishable from the clean ones, the testers will find it harder to decide an image is clean or poisoned. Hence, better backdoor methods led to higher fooling rates on not only backdoor inputs but also on clean ones. The rates from previous methods are low, with maximum 7.7% on all inputs, implying that they are obvious to humans to detect. In contrast, our rate is 28%, four times their best number. It confirms that WaNet is stealthy and hard to detect, even with trained people.

Although our backdoor images are natural-looking, some of them have subtle properties that can be detected by trained testers. We provide two of the most detected backdoor examples from WaNet in Fig. 6b. In the first case, the circle sign is not entirely round. In the second case, the right edge of the traffic sign is slightly curved. Although these conditions can be found on real-life traffic signs, they are not common in the testing dataset GTSRB. These images are of the minority, and our fooling rate on backdoor images is 38.6%, not far away from the rate of 50% in random selection.

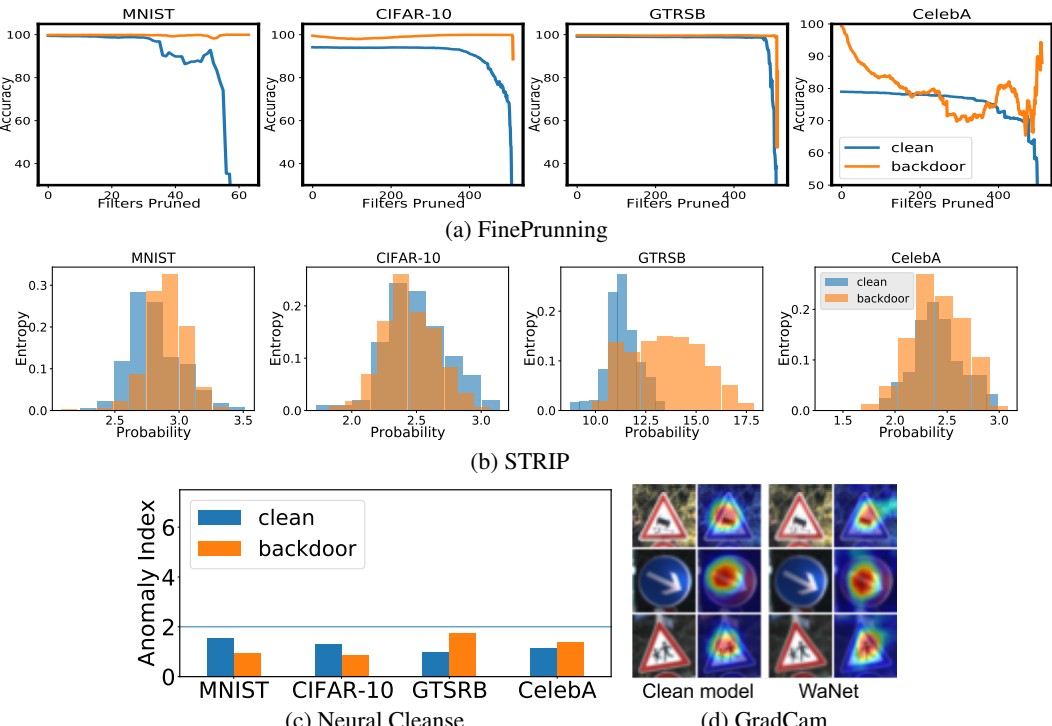

Figure 7: Experiments on verifying WaNet by the state-of-the-art defense and visualization methods.

## 4.4 DEFENSE EXPERIMENTS

We will now test the trained models against the popular backdoor defense mechanisms, including Neural Cleanse, Fine-Prunning (Model defenses), and STRIPS (Testing-time defense).

**Neural Cleanse (Wang et al., 2019)** is a model-defense method based on the pattern optimization approach. It assumes that the backdoor is patch-based. For each class label, Neural Cleanse computes the optimal patch pattern to convert any clean input to that target label. It then checks if any label has a significantly smaller pattern as a sign of backdoor. Neural Cleanse quantifies it by the Anomaly Index metric with the clean/backdoor threshold $\tau = 2$. We ran Neural Cleanse over our WaNet models and report the numbers in Fig. 7c. WaNet passed the test on all datasets; its scores are even smaller than the clean model ones on MNIST and CIFAR-10. We can explain it by the fact that our backdoor relies on warping, a different mechanism compared with patch-based blending.

**Fine-Pruning (Liu et al., 2018a)**, instead, focuses on neuron analyses. Given a specific layer, it analyzes the neuron responses on a set of clean images and detects the dormant neurons, assuming they are more likely to tie to the backdoor. These neurons are then gradually pruned to mitigate the backdoor. We tested Fine-Pruning on our models and plotting the network accuracy, either clean or attack, with respect to the number of neurons pruned in Fig. 7a. On all datasets, at no point is the clean accuracy considerably higher than the attack one, making backdoor mitigation impossible.

**STRIP (Gao et al., 2019)** is a representative of the testing-time defense approach. It examines the model with the presence of the input image. STRIP works by perturbing the input image through a set of clean images from different classes and raising the alarm if the prediction is persistent, indicating by low entropy. With WaNet, the perturbation operation of STRIP will modify the image content and break the backdoor warping if present. Hence, WaNet behaves like genuine models, with similar entropy ranges, as shown in Fig. 7b.

## 4.5 NETWORK INSPECTION

Visualization tools, such as GradCam (Selvaraju et al., 2017), are helpful in inspecting network behaviors. Patch-based backdoor methods can be exposed easily due to the use of small trigger

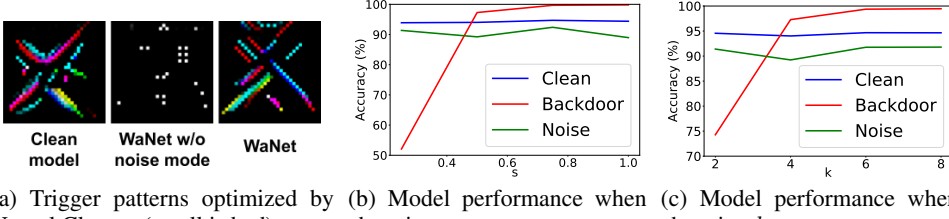

(a) Trigger patterns optimized by Neural Cleanse (small is bad)

(b) Model performance when changing $s$

(c) Model performance when changing $k$

Figure 8: Ablation studies on CIFAR-10 dataset: (a) Role of the noise mode training, (b,c) Network performance when changing warping hyper-parameters.

regions, as pointed out by Cheng et al. (2019); Doan et al. (2019). Our attack method is based on the warping on the entire image, so it is undetectable by this algorithm. We visualize activation based on the label that has the highest prediction score in Fig. 7d. With clean models, that label is for the correct class label. With WaNet and backdoor inputs, it is the backdoor label $\hat{c}$. As can be seen, the visualization heatmaps of WaNet look like the ones from any clean model.

## 4.6 Ablation Studies

**Role of the noise mode** Without the noise mode, we could still train a backdoor model with similar clean and attack accuracy. However, these models failed the defense test with Neural Cleanse as shown in Fig. 9, and the optimized trigger patterns revealed their true behavior.

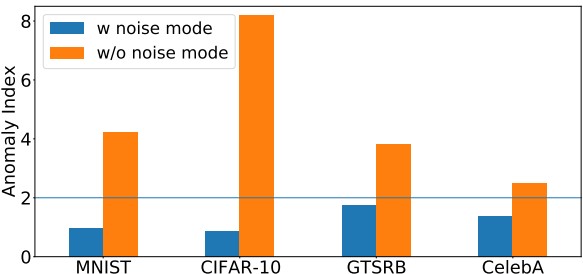

Figure 9: Networks' performance against Neural Cleanse with and without noise mode.

Fig. 8a displays the trigger patterns optimized by Neural Cleanse for the attacking class "airplane" on CIFAR-10. With the clean model, this pattern has an airplane-like shape, and it is big enough to rewrite image content given any input. With our model trained without noise mode, the optimized pattern just consists of scattered points. This pattern is remarkably smaller, making the model caught by Neural Cleanse. It reveals that the model did not learn the specific backdoor warping; instead, it remembered the pixel-wise artifacts. By adding the noise training mode, our model no longer relies on those artifacts, and the optimized pattern looks similar to the clean model's one.

**Other hyper-parameters** We investigated the effect of the warping hyper-parameters, including the strength $s$ and the grid size $k$. Fig. 8b and 8c show the clean, attack, and noise mode accuracy of our network on the CIFAR-10 dataset when changing each of these parameters. When $k$ or $s$ is small, the backdoor images are similar to the clean ones. However, since they are a minority ($\rho_a = 0.1$), the network would treat them like data with noisy labels in those scenarios. Hence, clean and noise accuracies are stable across configurations. In contrast, backdoor accuracy suffers on the left side of the plots. It gradually increases when $s$ or $k$ is small, then saturates and stays near 100%.

## 5 Conclusion and Future Works

This paper introduces a novel backdoor attack method that generates backdoor images via subtle image warping. The backdoor images are proved to be natural and undetectable by humans. We incorporate in training a novel "noise" mode, making it stealthy and pass all the known defense methods. It opens a new domain of attack mechanism and encourages future defense research.

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

# A APPENDIX

## A.1 SYSTEM DETAILS

### A.1.1 DATASETS

We used 3 standard datasets, from simple to more complex ones, to conduct our experiments. As the datasets are all used in previous related works, our results would be more comparable and reliable.

#### MNIST

The dataset (LeCun et al., 1998) is a subset of the larger dataset available from the National Institute of Technology (NIST). This dataset consists of 70,000 grayscale, $28 \times 28$ images, divided into a training set of 60,000 images and a test set of 10,000 images. Original dataset could be found at `http://yann.lecun.com/exdb/mnist/`.

We applied random cropping and random rotation as data augmentation for the training process. During the evaluation stage, no augmentation is applied.

#### CIFAR10

The dataset was introduced the first time by Krizhevsky et al. (2009). It is a labeled subset of the 80-millions-tiny-images dataset, collected by Alex Krizhevsky, Vinod Nair and Geoffrey Hinton, consists of 60,000 color images at the resolution of $32 \times 32$. The dataset contains 10 classes, with 6,000 images per one. It is divided into two subsets: a training set of 50,000 images and a test set of 10,000 images. The data set is public and avalable at `https://www.cs.toronto.edu/~kriz/cifar.html`.

During training stage, random crop, random rotation and random horizontal flip were applied as data augmentation. No augmentation was added at the evaluation stage.

#### GTSRB

The German Traffic Sign Recognition Benchmark - the GTSRB (Stallkamp et al. (2012)) is used as an official dataset for the challenge held at the International Joint Conference on Neural Network (IJCNN) 2011. This dataset consists of 60,000 images with 43 classes and the resolution varying from $32 \times 32$ to $250 \times 250$. It is divided into a training set of 39,209 images and a test set of 12,630. The dataset could be found at `http://benchmark.ini.rub.de/?section=gtsrb&subsection=dataset`.

Input images were all resized into $32 \times 32$ pixels, then applied random crop and random rotation at the training stage. No augmentation was used at the evaluation stage.

#### CelebA

CelebFaces Attributes Dataset - CelebA, first introduced by Liu et al. (2015), is a large-scale face attributes dataset. It contains 10,177 identities with 202,599 face images. Each image has an annotation of 5 landmark locations and 40 binary attributes. The dataset is publicly available at `http://mmlab.ie.cuhk.edu.hk/projects/CelebA.html`.

Noted that this dataset is highly unbalanced. Due to the time limitation, we select 3 out of 40 attributes, namely Heavy Makeup, Mouth Slightly Open and Smiling, as suggested by Salem et al. (2020). We then concatenate them into 8 classes to create a multiple label classification task. The input images were all resized into $64 \times 64$ pixels. Random crop and random rotation were applied as data augmentation at the training stage. No augmentation was applied at the evaluation stage.

### A.1.2 CLASSIFICATION NETWORKS

#### MNIST

We used a simple, self-defined structure as network classifier for this dataset. Detailed architecture will be mentioned in Table 2.

Table 2: Detailed architecture of MNIST classifier. $*$ means the layer is followed by a Dropout layer. $\dagger$ means the layer is followed by a BatchNormalization layer.

| Layer | Filter | Filter Size | Stride | Padding | Activation |
|-------|--------|-------------|--------|---------|------------|
| $Conv2d^{\dagger}$ | 32 | $3 \times 3$ | 2 | 1 | ReLU |
| $Conv2d^{\dagger}$ | 64 | $3 \times 3$ | 2 | 0 | ReLU |
| $Conv2d$ | 64 | $3 \times 3$ | 2 | 0 | ReLU |
| $Linear*$ | 512 | - | - | 0 | ReLU |
| $Linear$ | 10 | - | - | 0 | Softmax |

## CIFAR10 and GTSRB

For the CIFAR-10 and GTSRB datasets, we use PreActRes18 (He et al., 2016) architecture as classification networks.

## CelebA

For the CelebA dataset, we use ResNet18 (He et al., 2016) architecture as the classification network.

### A.1.3  RUNNING TIME

We use a system of a GPU RTX 2080Ti and a CPU i7 9700K to conduct our experiment. Detailed inference time of each module will be demonstrated below.

Table 3: Inference time of our modules.

| | MNIST | CIFAR10 | GTSRB | CelebA |
|---|-------|---------|-------|--------|
| $time/sample$ | $4.37\,\mu s$ | $18.64\,\mu s$ | $18.65\,\mu s$ | $87.51\,\mu s$ |

### A.2  ALL-TO-ALL ATTACK

Beside the single-target attack scenario, we also verified the effectiveness of WaNet in multi-target scenario, often called all-to-all attack. In this scenario, the input of class $y$ would be targeted into class $c(y) = (y + 1)\ mod\ |C|$, where $|C|$ is the number of classes.

### A.2.1  EXPERIMENTAL SETUP

We use the same experimental setups as in the single-target scenario, with a small modification. In the attack mode at training, we replace the fixed target label $\hat{c}$ by $(y + 1)\ mod\ |C|$. In the attack test at evaluation, we also change the expected label similarly.

### A.2.2  ATTACK EXPERIMENT

We conducted attack experiments and reported result in Table 4. While models still achieve state-of-the-art performance on clean data, the attack efficacies slightly decreases. This is due to the fact that the target label now varies from input to input. Though, the lowest attack accuracy is 78.58%, which is still harmful to real-life deployment.

Similar to all-to-one scenario, we also tested our model with noise mode and recorded the noise accuracy.

### A.2.3  DEFENSE EXPERIMENTS

We repeat the same defense experiments used in the all-to-one scenario. Our backdoor models could also pass all the tests mentioned in Figure 7.

Table 4: All-to-all attack result.

| Dataset | Clean | Attack | Noise |
|---------|-------|--------|-------|
| MNIST | 99.44 | 95.90 | 94.34 |
| CIFAR-10 | 94.43 | 93.36 | 91.47 |
| GTSRB | 99.39 | 98.31 | 98.96 |
| CelebA | 78.73 | 78.58 | 76.12 |

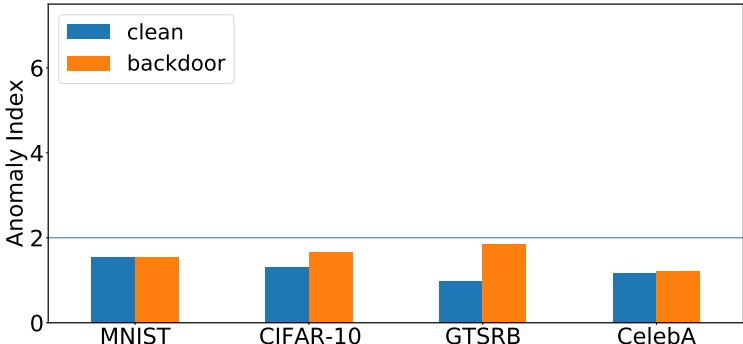

Figure 10: Neural Cleanse against all-to-all scenario.

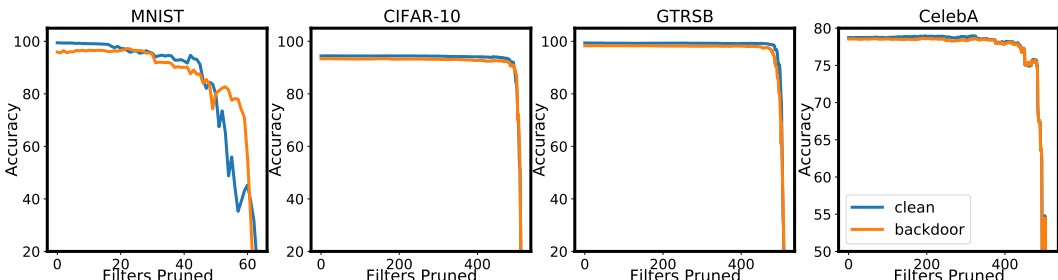

Figure 11: Fine-pruning against all-to-all scenario.

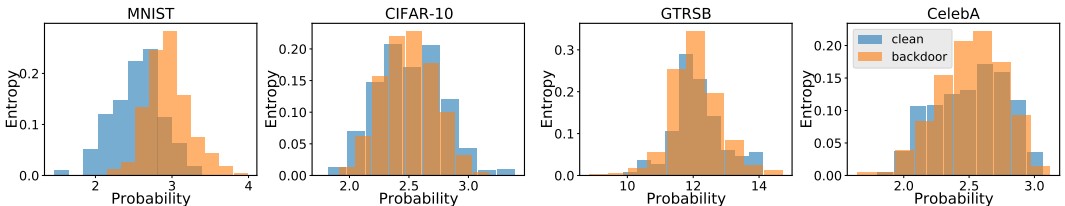

Figure 12: STRIP against all-to-all scenario.

## A.3  ADDITIONAL RESULTS

### A.3.1  ADDIONAL IMAGES FOR METIONED BACKDOOR ATTACK METHODS

We provide additional examples comparing backdoor images from WaNet and from other attack methods in Fig. 13.

### A.3.2  EXPERIMENT ON SPECTRAL SIGNATURE DEFENSE

Tran et al. (2018) proposed a data defense method based on the *spectral signature* of backdoor training data. Although this data-defense configuration does not match our threat model, we find it useful to verify if our backdoor data have the spectral signature discussed in that paper. We repeated the experiment in the last plot of its Fig. 1, using 5000 clean samples and 1172 backdoor samples

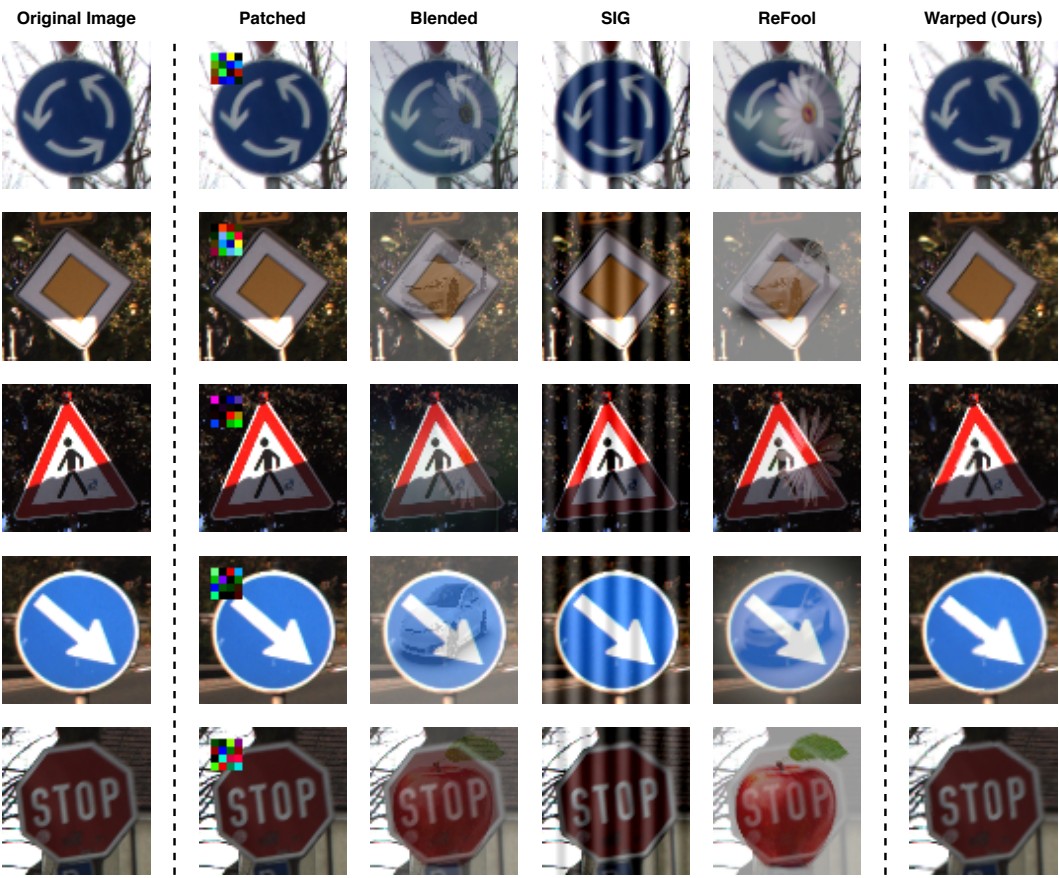

Figure 13: Additional images for mentioned backdoor attack methods.

generated by WaNet on the CIFAR-10 dataset, which is the same dataset used in the original paper. Fig. 14 plots histograms of the correlations between these samples' learned representations and their covariance matrix's top right singular vector. As can be seen, the histograms of the two populations are completely inseparable. Thereby, the backdoor training samples could not be removed from the training dataset using their proposed method. One possible explanation is that the distributional difference between the clean and backdoor correlations in the traditional backdoor methods was the result of the domination of a few backdoor neurons. We do not have such a phenomenon in WaNet, as proved in Fine-Prunning experiments, eliminating the appearance of spectral signature.

### A.3.3 THE STABILITY OF WANET

In this section, we verify if WaNet is stable to the variations of the warping field $M$. We trained 8 WaNet backdoor models, using 8 randomly generated warping fields, in the CIFAR10 dataset. The clean, backdoor, and noise accuracies of the trained models are all stable, as shown in Table 5.

Table 5: The stability of WaNet on the CIFAR-10 dataset.

|  | Clean | Backdoor | Noise |
|---|---|---|---|
| Accuracy (%) | $94.42 \pm 0.08$ | $99.40 \pm 0.21$ | $93.16 \pm 0.43$ |

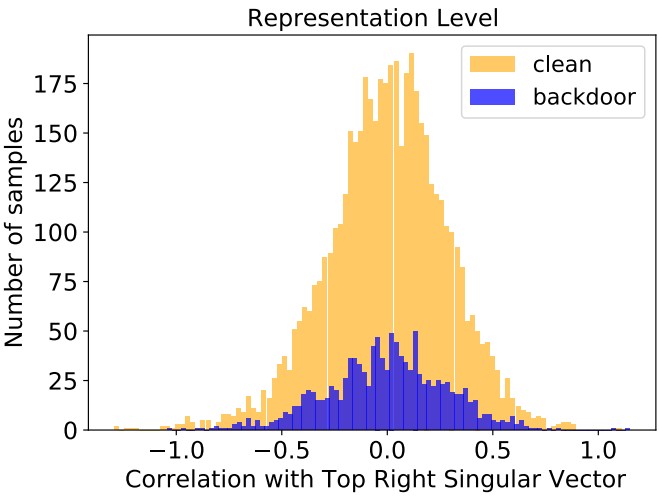

Figure 14: Spectral Signature

### A.3.4 ADDITIONAL TRIGGER PATTERNS VISUALIZING THE ROLE OF THE NOISE MODE

This section further demonstrates the importance of noise mode by providing trigger patterns optimized by Neural Cleanse on more datasets and with more target classes. Fig. 15a and 15b visualize the patterns on MNIST and GTSRB dataset using backdoor models trained for target label 0, similar to Fig. 8a. Fig. 15c, 15d, and 15e provide results on all three datasets but with backdoor models for label 3. As can be seen, the WaNet models without noise mode training return sparse and small patterns, thus easy to be detected by Neural Cleanse. By including that training mode, the optimized patterns are more crowded and approach clean models' ones. Note that we skip visualizing the results on the CelebA dataset; its patterns optimized on either clean or backdoor models are all too sparse and small for humans to analyze due to subtle differences between human faces.

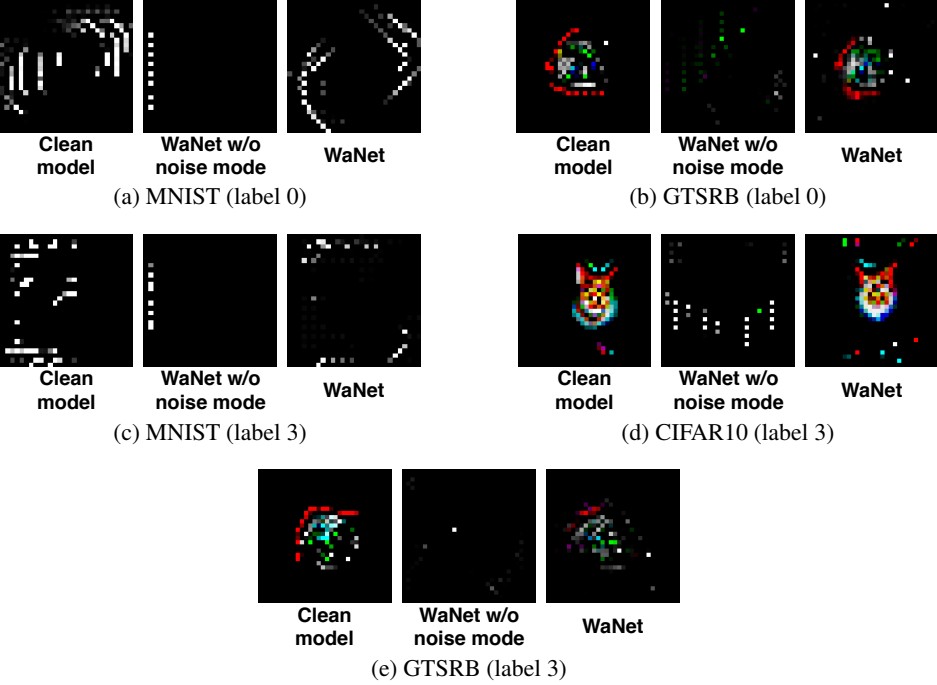

Figure 15: Additional trigger patterns optimized by Neural Cleanse for the target label (small is bad).

