# OpenReview forum: "WaNet - Imperceptible Warping-based Backdoor Attack"
_ICLR.cc/2021/Conference — ICLR 2021 Poster_

### Official Review · AnonReviewer4 · 2020-10-24
**A New Framework of Imperceptible Backdoor Attacks**

**Rating:** 7
**Confidence:** 4

**Review:**

This paper proposes a backdoor attack method using warp-based triggers, which distorts the global structure of the image (input) making the attack undetectable/un-noticeable by humans.
This global-distortion aspect of the attack makes effective against (besides humans) popular backdoor defense mechanisms such as neural cleanse and fine pruning.

Overall, I enjoyed reading this paper and the ideas proposed therein. However, I have some remarks about its clarity and some of the experiments that I hope the authors address.

-----

*Quality* (8/10):
The paper provides substantial experimental evidence that the attack method is effective in terms of:
1. Demonstrating the attack on a variety of datasets (MNIST, CIFAR10, GTSRB)
2. Ablation studies of the proposition components.

I'd have increased my Quality score if:
- The experiment of Ablation Studies (Role of Noise Mode) was presented with more supporting evidence rather than a single-class from a single dataset optimized trigger patterns (Figure 7a.)
- More datasets and/or various classifier architectures (Table 1)

-----

*Clarity* (6/10):
- There is room for improving the paper's write-up and presentation (e.g., some figures are missing their x-axis/y-axis labels as in Fig 6b).

-----
*Originality* (9/10):
I believe that this work is original. The bulk of backdoor attacks have been patch-based and this is one of few works that explore other backdoor alternatives.

-----
*Significance* (8/10):
This is a significant work and opens the door to new questions in the field of backdoor attacks.

-----
Pros:
- A new framework of imperceptible backdoor attacks that break away from the notion of patches  & water-marks.
-Thorough experiments that validate the effectiveness of the method.

Cons:
- Not enough supporting evidence to the role of Noise Mode in training.
- Some missing experiments (highlighted below) to further support the proposition.

----

**Remarks/Questions to the authors**:
- Fig 1: I suggest adding difference images for each of the methods, to highlight the differences each attack is incorporating into the image.
- Sec 2.4: The justification for the unsuitability of affine & projective transformations for attack purposes is not clear. Do you mean they are not going to be effective?
- Fig 2: Would be great to pinpoint some pixels of M and highlight what their values are in the [-1,1] interval besides the colormap.
- Sec 3.2: It appears that there are no constraints on the backward warping? A target pixel ,say at[-1,-1], can have a source pixel at [1,1]? It is not clear how the control grid (Eq. 3) ensures the locality of the warp. Please elaborate.
- Sec 3.2: Does M get generated once? or it's per image? I assume it is generated once.
- Sec 3.2: If there is a single M, how robust is the training to different realizations of M? The experiments do not discuss that.
- Table 2: Why "Clean Inputs" fooling rate differs across different attacks? Should not they be the same?
- Figure 6a: Please specify how clean mode is obtained (is it based on the first row of Eq. 6) while "backdoor" represents all of Eq. 6?
- Figure 6d: Please describe GradCam setup. Is it visualizing the activations based on the backdoor label $\hat{c}$?
- Figure 7b: I was expecting all the curves to start with a low-accuracy as the images from all the modes tend to be very similar for (k<6 and s<0.75; as shown in Figure 3) and so should not they suffer equally?

**Post-Rebuttal**
The authors have addressed my comments and revised the manuscript accordingly. Recommending an accept.

---

> ### Author Response · Authors · 2020-11-17
> **Thank for your detailed and constructing reviews. Here are our answers.**
>
> Thank for your detailed and constructing reviews. We truly appreciate that. We would clarify your concerns below and make changes in the revision version as your suggests.
>
> ## Quality:
> ### 1. More supporting evidences for the Ablation Studies (Role of Noise Mode).
> Thanks for your suggestions. We have added **Fig. 9** to provide more supporting evidences for that Ablation Studies. We show that without Noise Mode in training, the trained backdoor models will be easily detected by Neural Cleanse not only on CIFAR-10 but also on other datasets.
> ### 2. More testing datasets.
> We have added test results on **CelebA** dataset in the paper revision.
>
> ## Clarity:
> We have updated all x-axis/y-axis labels for all figures.
>
> ## Remarks/Questions:
> ###  1. Fig. 1: Adding differences images for each methods to highlight the differences each attack is incorporating into the image.
> Thanks for your great suggestion. We have added those images to Fig. 1 of the paper revision. We also have added various backdoor inputs for mentioned attack methods in the Appendix.
>
> ### 2. Sec. 2.4: The justification for the unsuitablity of affine and projective transforms for attack purposes is not clear.
> We meant they are unsuitable for attack purposes. Affine and projective transformations are naturally introduced to clean images via the image capturing process. If we apply these transformations to these images, the transformed images can be identical to other clean images that are of the same scenes but captured at different viewpoints. Hence, these transformations are not suitable to generate backdoor examples, particularly in physical attacks. We have updated our paper to make it clearer.
>
> ### 3. Fig. 2: Would be great to pinpoint some pixels of M and highlight  what their values are in the [-1,1] interval besides the colormap.
> We have added the lookup colormap to Fig. 2.
>
> ### 4. Elaborate the constraint on the backward warping.
> M is a warping "field", similar to motion field. It stores the relative position of the backward sampling points. For example, if the value of M at a pixel is [-1,-1], we need to sample at the "neighbor" pixel on the top-left. The average value of M is small and controlled by s, enforcing the locality of the warp.\
> We want to clarify that M is not the normalized sampling grid used in the API function grid_sample of PyTorch. Inside the warping function W, we do some processes to convert M to that sampling grid before inputting to grid_sample. We have updated our paper to emphasize it.
>
> ### 5. Sec. 3.2: Does M get generated once? or it's per image?
> Yes, M is generated once. We have highlighted it in the Threat Model section.
>
> ### 6. Sec. 3.2: If there is a single M, how robust is the training to different realizations of M?
> In case you asked about the stability of our training when using different values of M, it is pretty stable. For example, the accuracies in the attack experiments in Fig. 5a do not fluctuate over 0.5% compared to the reported numbers. We just used a single model for each dataset to have consistent results across experiments.
>
> ### 7. Table 2: Why "Clean Inputs" fooling rate differs across different attacks? Should not they be the same?
>  "Clean Inputs" fooling rate indicates the ratio of clean images flagged as backdoor. In each survey, the tester knew beforehand that half of the photos are backdoor. Hence, when the backdoor images were close to the clean ones, the tester would be less confident to identify an image as clean. Therefore, a better backdoor method led to a higher "Clean Inputs" fooling rate.
>
> ### 8. Fig. 6a: Please specify how clean mode is obtained (is it based on the first row of Eq. 6) while "backdoor" represents all of Eq. 6?
> In Fig. 6, "clean" means "clean model". The experiments show that our backdoor models are stealthy and behave like clean models.
> We can obtain clean models by training with only clean data (this means &#961;_a = &#961;_n = 0). Hence, you can say that it is based on the first row of Eq. 6.
>
> ### 9. Fig. 6d: Please describe GradCam setup. Is it visualizing the activations based on the backdoor label $\hat{c}$
> It is visualizing the activation based on the label that has the highest prediction score. With clean models, it is the correct class label. With WaNet and backdoor inputs, it is the backdoor label $\hat{c}$.
>
> ### 10. Fig. 7b: Why don't the curves suffer equally when s is small?
> We only use a small portion of training data as backdoor samples (**&#961;_a = 0.1**), similar to other backdoor attack methods. When k or s is small, the backdoor images are very similar to the clean ones. However, since they are a minority, the network will treat them like data with noisy labels. Therefore, the clean and noise-mode accuracy numbers are not affected, while the backdoor ones quickly drop.

---

> > ### Comment · AnonReviewer4 · 2020-11-18
> > **Thank you for your answers**
> >
> > The authors have adequately addressed most of my comments and revised the manuscript accordingly.
> >
> > Two main remarks:
> >
> > 1. For completeness and better clarity of the paper, it would be great to incorporate some of the answers that did not accompany any revisions in the manuscript into the paper.
> > E.g. Point 7, Point 9, and Point 10 in the authors' response above.
> >
> > 2. Point 6: a documented experiment to support "the robustness to different Ms" would be good to have in the paper.
> >
> > Thank you.

---

> > > ### Author Response · Authors · 2020-11-18
> > > **We have revised the paper as your suggestions.**
> > >
> > > Thanks a lot for your suggestions.
> > >
> > > We have added Point 7, Point 9, and Point 10 to the main text. We have also added a stability test to support "the robustness to different Ms" in the Appendix due to the page limit.
> > >
> > > Thank you.

---

> > > > ### Comment · AnonReviewer4 · 2020-11-18
> > > > **Increased my score. One last request.**
> > > >
> > > > Could you produce more trigger patterns like that of Figure 8a (for different datasets and classes) and put them in the appendix? This will be complementing the narrative of Figure 9.
> > > >
> > > > Thank you.

---

> > > > > ### Author Response · Authors · 2020-11-24
> > > > > **Thank you for your suggestion. We have added more trigger patterns.**
> > > > >
> > > > > Thanks a lot for your constructive comments and positive score. We have added more trigger patterns into the appendix.

---

### Official Review · AnonReviewer1 · 2020-10-27
**Interesting idea.**

**Rating:** 6
**Confidence:** 4

**Review:**

This paper proposes a new backdoor attack method on vision classifiers, where a warping-based trigger is adopted instead of patched triggers. To bypass backdoor detectors, the authors also propose a “noise” mode of training poisoned classifiers.
1. For the image warping, from Figure 3, I can not see any difference between most of the warped images and the original one. Also, the authors claim that WaNet is an imperceptible backdoor attack. However, Table 2 reported that the fooling rate of human on backdoor inputs and clean inputs are 38.6% and 17.4% respectively, this suggests that there may still be some differences between backdoor inputs and clean inputs that can be identified by human. Can the authors show some such examples?
2. I understand that the backdoor in this paper is the unique warping function. I think the authors should explicitly mention this as part of the threat model of WaNet.
3. The authors mentioned in section 3.2 that “To get such a warping field, we extend the idea of TPS with some modifications”. What is the extension and what is the difference from the original TPS? I think the authors should explicitly define their contributions while describing the technical details.
4. Did the authors try the defense “Spectral signatures in backdoor attacks” (Tran et al. 2018)? I think this is a valid defense method worth evaluating for WaNet.
5. I suggest the authors briefly describe how the image warping process is conducted. Currently it is not discussed in the paper, which may not be good for interested readers.

Overall i am satisfied with the paper presentation. The idea is interesting and the paper is self-contained. I would like the authors to address the issues I raise.

---

> ### Author Response · Authors · 2020-11-17
> **Thank you for your insigntful comments. Here are our answers.**
>
> Thank you for your insigntful comments. We would address your concerns below:
>
> ### 1. Examples in which differences between backdoor inputs and clean inputs that can be identified by human:
> Although our backdoor images are natural-looking, some of them have subtle properties that can be detected by trained testers. In some cases, a straight line on the edge of the traffic sign is slightly curved. In other instances, the circle on the sign is not perfectly round. Although these conditions can be found on real-life traffic signs, they are not common in the testing dataset GTSRB. These images are of the minority, and our fooling rate on backdoor images is 38.6\%, not far away from the rate of 50\% in random selection. We have added these examples and analyses in the paper revision.
>
> ### 2. Should explicitly mention the warping function as part of the threat model of WaNet:
> Thanks for your suggestion. We have updated our paper accordingly.
>
> ### 3. Should explicitly mention the extension and the difference from the original TPS:
> Thanks for your suggestion. We have revised our paper to make it clearer. We borrow from TPS the idea of using control points and warping field interpolation. However, we simplify it in our system for more efficient implementation. Given control points in a uniform grid, we do not use the complex spline-based interpolation but a simple bicubic one.
>
> ### 4. Try the Spectral signature defense method:
> This defense method is data-defense; it assumes that the defender has control over the training process, and the adversary attacks by providing infected training data. This assumption, however, does not match our threat model, where the already-trained backdoor model is provided by a third party.\
> Nonetheless, we did verify if our backdoor data have the spectral signature discussed in that paper. We repeated the experiment in the last plot of their Fig. 1, using 5000 clean samples and 1172 backdoor samples generated by WaNet on the CIFAR-10 dataset (the same dataset of the original paper). The histograms of the two correlation populations were inseparable. Thereby, the backdoor samples could not be removed from the training dataset using their proposed method. Due to the main paper's page limit, we had added this experiment in the Appendix.
>
> ### 5. Should briefly describe how the image warping process is conducted:
> We have updated our paper to describe it clearer in page 4, after Equation (2). After having the warping field M, we use it convert any clean input image to a backdoor one using a warping function W. M is motion-field-like, which defines the relative sampling location for backward warping at each point in the target image. W allows a floating-point warping field as input. When a sampling pixel falls on non-integer 2D coordinates, it will be bi-linear interpolated. To implement W, we rely on the grid_sample API provided by PyTorch. This API, however, inputs a grid of normalized absolute coordinates of each sampling point. To use that API, we first sum M with an identity sampling grid, then normalize to [-1, 1] to get the required grid input.

---

> > ### Comment · AnonReviewer1 · 2020-11-18
> > **One question on Spectral Signature**
> >
> > The authors have addressed most of my concerns. I have one question though. Is there an intuition why your proposed attack is
> >  immune to the Spectral Signature defense?

---

> > > ### Author Response · Authors · 2020-11-18
> > > **Our possible explanation.**
> > >
> > > Thanks a lot for your feedback.
> > >
> > > Regarding your question, one possible explanation is that the distributional difference between the clean and backdoor correlations in the traditional backdoor methods was the result of the domination of a few backdoor neurons. We do not have such a phenomenon in WaNet, as proved in Fine-Prunning experiments, eliminating the appearance of spectral signature.

---

> > > > ### Comment · AnonReviewer1 · 2020-11-18
> > > > **Thanks for your response**
> > > >
> > > > Thanks for the explanation. I have increased my score.

---

### Official Review · AnonReviewer2 · 2020-10-28
**Review for WaNet**

**Rating:** 6
**Confidence:** 2

**Review:**

1. Summary
- This paper presents the imperceptible warping based back-door attack technique.
- With the poisoned image generated from the technique, the network will be fooled.

2. Strong points
- The technique produces the imperceptible poisoned image.
- Looks superior to other techniques.

3. Weak points
- I am not sure that the experiments are enough to show the technique's superiority.
- Why did you perform experiments only on small datasets such as CIFAR-10, MNIST, and GTSRB?


I am not confident in the review because I am not familiar with the attack algorithms.
I can conclude my decision with other reviewer's reviews.

---

> ### Author Response · Authors · 2020-11-17
> **Thank you for your kind response. Here are our answers.**
>
> Thank you for your kind response. We hope that our work would make you get into this backdoor topic. We would address your concerns below:
> ### 1. The sufficiency of experiments to show the technique's superiority:
> We did a profound list of experiments to prove the advantages of our attack method:
> - Our method showed very high accuracy on attack experiments like reported in other backdoor attack papers.
> - This paper is one of very few works that conduct detailed defense experiments in both Model Defense and Testing-time defense scenarios. Our methods passed all these defense mechanisms.
> - We demonstrate the stealthiness of this method against the network inspector (GradCAM), which has just been proposed recently as an effective method for backdoor detection.
> - We also demonstrate the stealthiness of this method via Human evaluation. To our best knowledge, no backdoor attack paper has done it before.
> - Besides, we illustrate the effectiveness of our backdoor inputs when recaptured in the wild. It guarantees real-life applications of this method.
>
> ### 2. Why did we conduct experiments only on small datasets namely CIFAR-10, MNIST and GTSRB:
> These datasets are widely used in the previous backdoor attack and backdoor defense papers. Conducting experiments on these datasets allows us to fairly compare to the earlier attack methods and testify on existing defense mechanisms.
> Nonetheless, we have added experiments on **CelebA**, a relatively large dataset with 202,599  images in the paper revision.

---

### Decision · Program_Chairs · 2021-01-07
**Final Decision**

**Decision:**

Accept (Poster)

**Comment:**

The paper presents a new method for generation of backdoor attacks against deep networks. The new method uses global warping instead of noise patches which makes the attack much more stealthy than previous approaches. The attack effectiveness is demonstrated on 3 benchmark datasets. A small user study is carried out to demonstrate that the attack is stealthier than conventional backdoor attacks.

The new attack is a novel and original contribution which is likely to advance the understanding of backdoor attacks. There were some issues with respect to clarity in the original manuscript but the authors adequately addressed the critical remarks raised by the reviewers.